# Simultaneous Determination of Twenty Mycotoxins in the Korean Soybean Paste Doenjang by LC-MS/MS with Immunoaffinity Cleanup

**DOI:** 10.3390/toxins11100594

**Published:** 2019-10-12

**Authors:** So Young Woo, So Young Ryu, Fei Tian, Sang Yoo Lee, Su Been Park, Hyang Sook Chun

**Affiliations:** Advanced Food Safety Research Group, BK21 Plus, School of Food Science and Technology, Chung-Ang University, Anseong 17546, Korea; mochalatte9@naver.com (S.Y.W.); kfbthdud123@gmail.com (S.Y.R.); tianfei_real@163.com (F.T.); dm3822@naver.com (S.Y.L.); sirius6100@naver.com (S.B.P.)

**Keywords:** mycotoxins, *doenjang*, simultaneous determination

## Abstract

Doenjang, a Korean fermented soybean paste, is vulnerable to contamination by mycotoxins because it is directly exposed to environmental microbiota during fermentation. A method that simultaneously determines 20 mycotoxins in doenjang, including aflatoxins (AFs), ochratoxin A (OTA), zearalenone (ZEN), and fumonisins (FBs) with an immunoaffinity column cleanup was optimized and validated in doenjang using LC-MS/MS. The method showed good performance in the analysis of 20 mycotoxins in doenjang with good linearity (*R*^2^ > 0.999), intra- and inter-day precision (<16%), recovery (72–112%), matrix effect (87–104%), and measurement uncertainty (<42%). The validated method was applied to investigate mycotoxin contamination levels in commercial and homemade doenjang. The mycotoxins that frequently contaminated doenjang were AFs, OTA, ZEN, and FBs and the average contamination level and number of co-occurring mycotoxins in homemade doenjang were higher than those in commercially produced doenjang.

## 1. Introduction

Mycotoxins, produced mainly by the genera *Aspergillus, Penicillium, Fusarium, Claviceps*, and *Alternaria* [1], are unintentional contaminants of agricultural products and food; they cause hepatotoxicity, kidney toxicity, reproductive toxicity, and economic loss [2]. Therefore, most countries regulate the levels of the following mycotoxins in food and feeds: aflatoxin B1 (AFB1), aflatoxin B2 (AFB2), aflatoxin G1 (AFG1), aflatoxin G2 (AFG2), aflatoxin M1 (AFM1), ochratoxin A (OTA), zearalenone (ZEN), T-2 toxin, HT-2 toxin, deoxynivalenol (DON), fumonisin B1 (FB1), fumonisin B2 (FB2), and fumonisin B3 (FB3).

Doenjang, a fermented soybean paste produced in Korea, is a highly nutritious protein source with a high storage stability. During fermentation, the microbiota present in the surrounding environment naturally inoculate the paste and participate in its fermentation. However, simultaneously, this process is vulnerable to contamination by mycotoxigenic fungi that produce aflatoxins (AFs), such as *Aspergillus flavus*.

The main challenges in accurately determining AF levels in doenjang are matrix effects, such as adsorption and interference, and low AF recoveries. Doenjang contains large quantities of salt (7–23%), high levels of protein and brown pigments, and molecular fermentation by-products. These matrix components have the potential to interfere with the analysis of AFs [3]. Studies to determine mycotoxin levels in doenjang have mainly been conducted to detect AFs. However, to date, very few studies have focused on other mycotoxin contaminants in doenjang, including OTA.

Most methods used for the extraction and purification of mycotoxins in doenjang are solid-phase extractions using an immunoaffinity column (IAC) cleanup. For the quantification of trace levels of mycotoxin in soybean paste matrix, analysis without the clean-up process or using solid phase extraction (SPE) that was packed with general C18 or ion pair materials resulted in insufficient removal of salts and other polar components in the matrix [4]. Analyses of mycotoxins in doenjang, particularly AFs, are typically conducted using high-performance liquid chromatography coupled with fluorescence detection (HPLC-FLD). However, because reverse-phase eluents quench the fluorescence of AFB1 and AFG1 [5], chemical derivatization is frequently required to detect these analytes. Pre-column or post-column derivatization is thus performed with a suitable fluorophore to improve detectability. In addition, the AFs in doenjang have been analyzed by ELISA and LC-MS/MS [4,6,7,8,9,10]. Although HPLC-FLD is more affordable and easier to maintain than LC-MS/MS, LC-MS/MS analysis is more sensitive and suitable for simultaneous determination because it can analyze the mycotoxins lacking fluorophore groups, such as trichothecenes. To the best of the authors’ knowledge, there are no reported studies on the simultaneous determination of multiple mycotoxins in doenjang.

Currently, the risk management of mycotoxins in food is assessed using a single mycotoxin [11]. There are, however, various mycotoxins that may contaminate food simultaneously; humans are therefore exposed to the synergistic effect of multiple mycotoxins [12,13,14]. Thus, it is important to establish an accurate and reliable analytical method for the simultaneous determination of multiple mycotoxins. HPLC-MS has frequently been used to simultaneously determine various mycotoxins in food, particularly by LC coupled with triple quadrupole (QqQ)-MS using electrospray ionization (ESI) mode. QqQ-MS operating in multiple-reaction monitoring mode (also called selective reaction monitoring, SRM mode) provides a more sensitive analysis of targeted or known mycotoxins when using one precursor ion and a few product ions for each analyte compared with full spectral data [15,16]. Furthermore, QqQ-MS analysis can reduce dwell times, with faster data collection, without reducing the signal-to-noise ratio of the chromatographic peak, which also meets the requirements of UPLC, a recent replacement of HPLC [17]. ESI as an ionization source can be affected by signal suppression or enhancement, but this can be improved, at least partially, by appropriate sample preparation (cleanup), sample dilution, matrix-matched calibration, and isotope dilution mass spectrometry techniques [17].

A cleanup process is important to improve the selectivity and matrix effects by removing interference from the complex food matrices [18]. In this regard, the application of IAC in mycotoxin analysis has been a widely used method in recent years, and a fast, easy, inexpensive, effective, robust and safe (QuEChERS) strategy is also used because of its high selectivity and ability to reduce matrix effects [19]. Recently, multifunctional IACs have been used for the simultaneous analysis of AFs, OTA, ZEN, T-2/HT-2, DON, and FBs. These multifunctional IACs are commonly used to determine biomarkers of exposure to mycotoxins [20,21,22]. Most studies of food matrixes have been conducted on corn, cereal, and their products, and analyses of 11–14 mycotoxins have been reported [11,23,24,25,26]. Some methods require two time-consuming steps to extract mycotoxins from samples [23,25]. Another method uses water to elute mycotoxins from IACs, but this requires longer procedure times because of the need to evaporate the water under a stream of N_2_. These procedures are also laborious [24].

Therefore, the objective of the present study was to optimize and validate an efficient LC-MS/MS-based method with multifunctional IAC cleanup that can simultaneously determine 20 mycotoxins. The validated method was then to be applied to analyze multiple mycotoxins in samples of commercial and homemade doenjang.

## 2. Results and Discussion

### 2.1. Optimization of LC-MS/MS Parameters

Among the 20 tested mycotoxins, the analysis of 17 was optimized in the positive ESI mode and three in negative mode based on the signal intensity of the precursor ion. Most mycotoxins were ionized in the positive mode in the form of [M + H]^+^ ions. Some of the ZEN metabolites showed higher intensities as [M + H − H_2_O]^+^. The responses of [M + Na] ^+^ ions generated from T-2 and HT-2 toxins were higher than those of the other molecular forms, and this result is consistent with previous studies [27,28,29]. Nivalenol (NIV), DON, and 3-acetyl deoxynivalenol (3ADON) were optimized as the formic acid adduct [M + HCOO^–^]^–^ in negative ionization mode. For two product ions of each mycotoxin, a quantifier and qualifier were selected based on the signal intensity. The optimized parameters are tabulated in Table 1, and the LC-MS/MS chromatograms of the analytes are shown in Figure 1. 

### 2.2. Optimization of Sample Preparation Method

Twenty-three mycotoxins (AFB1, AFB2, AFG1, AFG2, sterigmatocystin (STG), OTA, ochratoxin B (OTB), ZEN, zearalanone (ZAN), α-zearalenol (α-ZEL), α-zearalanol (α-ZAL), β-zearalenol (β-ZEL), β-zearalanol (β-ZAL), T-2, HT-2, NIV, DON, 3ADON, 15-acetyl deoxynivalenol (15ADON), deoxynivalenol-3-glucoside (D3G), FB1, FB2, and FB3) that might cross-react with antibodies used for the IAC were tested. A recovery test was conducted by spiking a blank doenjang sample with each mycotoxin at 10 μg/kg for AFs and OTA, and 100 μg/kg for other mycotoxins. Furthermore, SSE (%) was determined to consider co-extraction of the analyte/matrix combination. The effects of the various cleanup methods on recoveries and matrix effects are shown in Figure 2.

#### 2.2.1. SPE Method (Method A)

Using method A, an Isolute Myco column was able to purify the following 19 mycotoxins from the doenjang matrix: AFB1, AFB2, AFG2, STG, OTA, OTB, ZEN, ZAN, *α*-ZEL, α-ZAL, *β*-ZEL, *β*-ZAL, T-2 toxin, HT-2 toxin, 3ADON, D3G, FB1, FB2, and FB3. The Isolute Myco column was not suitable for the analysis of type B trichothecenes (NIV, DON, 3ADON, 15ADON, and D3G); recoveries were poor, ranging from N.D. to 10.2%. On the other hand, method A yielded successful recoveries of FBs, ranging from 90.4% to 100.7%. In general, an SSE (%) of 80–120% is considered acceptable. Values outside this range indicate a strong matrix effect [30,31]. In terms of this SSE range, more than half of the tested analytes were highly suppressed (SSE < 80%), while 21.7% of the mycotoxins were not suppressed or enhanced using this method.

#### 2.2.2. QuEChERS (Method B)

Method B was able to analyze the following 15 mycotoxins: STG, OTB, ZEN, ZAN, *α*-ZEL, *β*-ZEL, T-2, HT-2, NIV, DON, 3ADON, D3G, FB1, FB2, and FB3. Method B showed satisfactory recoveries of type A trichothecenes (T-2 and HT-2), ranging from 78.3% to 82.6%. However, the AFs and OTA peak responses could not be detected because of interference peaks. The QuEChERS sorbents did not effectively remove pigments and debris such as protein from the matrix. Method B also showed signal suppression (SSE < 80%) of more than half of the mycotoxins. Only 13% of the analytes were not affected by the matrix.

#### 2.2.3. IAC Cleanup (Method C)

The IAC showed cross-reactivity with 22 mycotoxins. OTB did not cross-react with antibodies in the column. STG and 15ADON showed recoveries of <40% because they had less cross-reactivity with the antibodies in the IAC. FBs, which are the most water-soluble compounds, showed lower recoveries (<60%) than the other mycotoxins. Method C afforded a 60–120% recovery of all mycotoxins, except for STG, OTB, 15ADON, and FBs. Of the tested mycotoxins, 60.9% were not affected by the matrix. Furthermore, matrix-induced signal enhancement was observed in 34.8% of the analytes. Among them, FBs were the most enhanced, with %SSE ranging from 144.3% to 172.5%.

Method C was better overall in terms of compounds interfering with chromatographic peaks, recovery, precision, and the number of purified mycotoxins. Therefore, further optimization using the IAC method for the simultaneous determination of mycotoxins was conducted, but excluding STG, OTB, and 15ADON, which had no or weak cross-reactions with antibodies in the IAC.

The optimized method here was as follows. In the extraction step, NaCl (1 g) was added to reduce the interference from polar compounds in the food matrix and prevent the formation of an emulsion. In this study, the two-stage extraction procedure using water (40 mL) and subsequent MeOH (60 mL) was found to be efficient because the 20 mycotoxins considered here had a wide range of polarities (logP values ranging from −0.81 to 3.66). After the extraction and subsequent evaporation of the organic solvent phase, the loading of the resulting extract onto the IAC without filtration was time consuming; a significant amount of debris remained in the extract. Therefore, before loading onto the IAC, the extract was diluted four-fold with PBS, filtered through a Whatman GF/A filter to eliminate debris, and then loaded onto the column. The washing step was first carried out with 10 mL of water. However, this resulted in 20.5–26.5% loss of FBs, which are water soluble analytes (data not shown). Accordingly, the washing volume was reduced from 10 mL to 4 mL to minimize the loss of FBs in the washing step. The elution of analytes from the IAC with a mixture of MeOH and water is time consuming because water is slow to evaporate under N_2_. Therefore, water as the elution solvent was replaced with MeOH containing 0.2% formic acid. In the subsequent procedure, MeOH containing 0.2% formic acid was used, which increased the elution efficiency of the FBs.

### 2.3. Method Validation

#### 2.3.1. Linearity, LOD, LOQ, and Matrix Effect

Linearity was evaluated by the coefficient of determination (*R*^2^) of the matrix-matched external calibration curve, although *R^2^* is an indicator of goodness-of-fit rather than linearity. The calibration curves of six points in the ranges 0.5–20 μg/kg (for AFs and OTA) and 25–1000 μg/kg (for other mycotoxins) showed excellent linearities (*R*^2^ > 0.9994). The LOD and LOQ were calculated as 0.06–4.68 μg/kg and 0.17–14.18 μg/kg, respectively (Table 2). These values are lower than those of other reported LC-MS/MS methods for multiple mycotoxins in various matrices [32,33,34].

The matrix effect can cause signal suppression or enhancement of the analytes and affect the ion intensity, related to recovery and repeatability. The SSE effects of all mycotoxins were in the range 86.5–103.8% (Table 2). Matrix effects were improved to within an acceptable range through the optimized method C.

#### 2.3.2. Recovery, Precision, and Trueness

The mycotoxins were classified into two groups based on the similarity of their LOQ levels. Group 1 included AFs and OTA; their spiking levels were 1 μg/kg (2 × LOQ), 2 μg/kg (5 × LOQ), 3 μg/kg (10 × LOQ), and 6 μg/kg (20 × LOQ). Group 2 included other mycotoxins besides AFs and OTA; their spiking levels were 15 μg/kg (2 × LOQ), 37 μg/kg (5 × LOQ), 74 μg/kg (10 × LOQ), and 148 μg/kg (20 × LOQ). Each value was rounded to the first decimal place. The intra- and inter-day validation results are tabulated in Table 3. The mean accuracies for each mycotoxin were 96.2% for AFs, 78.8% for OTA, 88.7% for ZEN and ZEN metabolites, 83.5% for T-2 and HT-2, 94.1% for type B trichothecenes (NIV, DON, 3ADON, and D3G), and 91.3% for FBs. All tested recovery and precision values for AFB1, AFB2, AFG1, AFG2, OTA, DON, ZEN, T-2, HT-2, FB1, and FB2 were acceptable within the guidelines of the Commission Regulation (EU) No. 519/2014 [35]. The trueness for OTA was calculated as –0.01%, which is acceptable according to the guidelines presented by the European Commission Decision 2002/657/EC: a range of –20% to +10%, if the mass fraction is 10 μg/kg [36].

The recovery and precision of this method are similar to, or better than, the previously reported LC-MS/MS methods for multiple mycotoxins in various food matrices [26,37,38].

#### 2.3.3. Measurement Uncertainty

The expanded uncertainty was calculated for each mycotoxin at 2×, 5×, 10×, and 20 × LOQ spiking levels. The mean uncertainties were 15.9% for AFs, 35.6% for OTA, 19.7% for ZEN and ZEN metabolites, 21.4% for T-2 and HT-2, 11.7% for type B trichothecenes, and 12.2% for FBs (Table 4). The EC guidelines suggest that if the sample concentration is <100 μg/kg, then the acceptable expanded uncertainty is within 44%, and if the concentration is >100 μg/kg and <1000 μg/kg, then the acceptable expanded uncertainty is within 32% [39]. The uncertainties at 2×, 5×, 10×, and 20 × LOQ of all the mycotoxins are acceptable according to the criteria within the EU guidelines.

The contributions of each of the factors to the total uncertainty are shown in Appendix A. The most significant contribution to the total uncertainty was from the matrix (recovery and precision of the method), except for AFs, ZAN, α-ZEL, β-ZEL, and β-ZAL, which were most influenced by the standard preparation. Uncertainties from the standard preparation of AFs were evaluated to be relatively high because the accuracies were close to 100%; thus, the uncertainties from the matrix were low. Furthermore, ZEN derivatives were prepared using neat chemicals to obtain standard solutions. Therefore, the uncertainties from the standard preparation of ZEN derivatives included additional uncertainties from apparatus used: the balance and pipette.

### 2.4. Application to Actual Samples

A total of 60 samples of commercial (*n* = 30) and homemade (*n* = 30) doenjang were analyzed using the optimized method. At least one mycotoxin contaminated 81.7% (49/60) of the samples. The overall results of the mycotoxin contamination levels are summarized in Table 5.

The most frequently contaminating mycotoxins in both commercial and homemade doenjang were FB1 and FB2, followed by OTA, AFB1, ZEN, and FB3. The mycotoxins with the lowest incidence were T-2 and 3ADON (not detected in all samples). Of the 60 samples, 35.0% (21/60) were contaminated with at least one AF within the range 0.11–5.43 μg/kg, 38.3% (23/60) contained OTA within the range 0.16–23.27 μg/kg, 53.3% (32/60) contained ZEN and its derivatives within the range 4.67–95.08 μg/kg, 20.0% (12/60) contained trichothecenes within the range 1.69–26.52 μg/kg, and 50.0% (30/60) contained at least one FB within the range 2.48–68.52 μg/kg. The OTA level (23.27 μg/kg) of one commercial product from the traditional market exceeded the regulatory limit of the Korean Government (>20 μg/kg not permitted). The contamination frequency of FBs was high, but the contamination level was low, and a greater diversity of mycotoxins was detected in homemade doenjang than in commercial products. This is probably because of the variety of raw materials used in the manufacture of homemade products, and thus, the origins of mycotoxins also varied. Furthermore, some mycotoxins occurred at a higher frequency or level in homemade doenjang than in commercial products.

To determine the differences in contamination levels between commercial and homemade doenjang, a Student’s *t*-test was performed (*p* < 0.05 and *p* < 0.01). The contamination levels of AFB1, AFG1, ZEN, ZAN, β-ZEL, FB1, and FB2 were significantly higher in homemade doenjang compared with commercial products. This result is similar to previous findings, namely that AF is present at higher levels in homemade doenjang than in commercial samples [4,6,7,8,9,10]. Interestingly, 63.3% (38/60) of the samples were cocontaminated with at least two mycotoxins. The percentage of samples cocontaminated with mycotoxins in commercial and homemade doenjang is presented in Figure 3. The commercial doenjang was contaminated with up to six mycotoxins, but the contamination levels were low: 0.27 µg/kg AFB1, 7.67 µg/kg NIV, 4.78 µg/kg DON, 4.05 µg/kg FB1, 5.78 µg/kg FB2, and 6.10 µg/kg FB3. In homemade doenjang, two samples were contaminated with nine mycotoxins. One sample was contaminated with 2.27 µg/kg AFB1, 0.96 µg/kg AFB2, 0.57 µg/kg AFG1, 3.99 µg/kg OTA, 38.10 µg/kg ZEN, 7.18 µg/kg α-ZAL, 25.70 µg/kg β-ZAL, 5.52 µg/kg FB1, and 6.71 µg/kg FB2, and the other sample was contaminated with 0.34 µg/kg AFB1, 19.68 µg/kg OTA, 75.63 µg/kg ZEN, 7.11 μg/kg ZAN, 4.96 μg/kg α-ZAL, 7.38 μg/kg β-ZEL, 11.34 μg/kg FB1, 6.80 μg/kg FB2, and 2.90 μg/kg FB3. The number of contaminating mycotoxins in homemade doenjang was higher than that in commercial doenjang.

The co-occurrence between mycotoxins is summarized in Figure 3. The co-occurrence of the fumonisin B series occurred the most frequently in commercial doenjang, and OTA–ZEN and DON–FB1 were the second most frequently co-occurring mycotoxins. In homemade doenjang, the co-occurrence of FB1–FB2, OTA–FB2, and OTA–ZEN was frequently observed. In particular, 40% (2/5) of AFB1-positive commercial samples and 64% (11/17) of AFB1-positive homemade samples were cocontaminated with OTA, and 60% (3/5) of OTA-positive commercial samples and 67% (12/18) of OTA-positive homemade samples were cocontaminated with ZEN. Greater cocontamination of mycotoxins occurred in the homemade doenjang than in the commercial doenjang.

This difference is attributed to the different production processes of commercial and homemade doenjang. Commercial products are fermented by artificial inoculation using a specific strain (e.g., *Aspergillus oryzae* or *Aspergillus sojae*), but homemade doenjang is naturally fermented with ambient microorganisms [40]. Thus, the homemade process is vulnerable to contamination by mycotoxigenic fungi.

In conclusion, an efficient LC-MS/MS-based method with multifunctional IAC cleanup was successfully developed, and validated, for the simultaneous determination of 20 mycotoxins. This method can be applied to other fermented soybean foods with similar matrices, such as Japanese miso, Indonesian tempeh, and Chinese sufu (furu). Furthermore, to the best of our knowledge, this study is the first attempt to simultaneously determine up to 20 mycotoxins after sample cleanup using multifunctional IACs in soybean fermented foods. In the doenjang samples collected in this study (a rather limited number of samples), 81.7% of the samples were contaminated with at least one mycotoxin and 1.7% exceeded the present regulatory limits for OTA. It is recommended that further monitoring with a larger number of samples be conducted for the sake of public safety regarding mycotoxins in doenjang.

## 3. Materials and Methods 

### 3.1. Chemicals and Reagents

Certified mycotoxin standards of AFB1, AFB2, AFG1, AFG2, STG, OTA, OTB, ZEN, T-2, HT-2, NIV, DON, 3ADON), 15ADON, and D3G were purchased from Romer Labs (Union, MO) and dissolved in acetonitrile (ACN). Certified standard solutions of FB1, FB2, and FB3 in ACN/water (50:50, *v*/*v*) were also purchased from Romer Labs. ZEN metabolites (ZAN, α-ZEL, α-ZAL, β-ZEL, and β-ZAL) were obtained from Sigma-Aldrich (St. Louis, MO) and prepared in ACN. FB1, FB2, and FB3 were stored at 4 °C. Other standard solutions were stored at –20 °C. A certified reference material (CRM) of doenjang powder for OTA (49.50 ± 1.17 µg/kg) was purchased from the Korea Research Institute of Standards and Science (KRISS) and used for the trueness test.

### 3.2. Sampling

We collected 60 samples (30 commercial and 30 homemade) of doenjang in 2018–2019. Commercial doenjang samples were purchased from online retailers, supermarkets, and local markets. Homemade doenjang samples were collected from four areas in Korea (Gyeonggi, Chungcheong, Jeolla, and Gyeongsang) using the snowball method, which is a nonprobability sampling technique in which existing study subjects recruit future subjects from among their acquaintances [41]. Collected samples were stored at 4 °C and then equilibrated at room temperature prior to analysis.

### 3.3. Sample Preparation

#### 3.3.1. Method A

In the solid phase extraction (SPE) method (Method A), an Isolute (Myco, 3 mL, 60 mg) column (Biotage, Sweden) was selected for use, because it employs a simple SPE procedure to bind mycotoxins of interest selectively to the sorbent. Sample preparation was conducted according to a multimycotoxin analysis method for vegetable raw materials and their processed products, as specified by the Korean Food Code (2018) and operator’s application note, with some modifications [42]. Briefly, 5 g of sample was extracted with 20 mL of 50% ACN (0.1% formic acid, *v*/*v*) by shaking for 30 min, followed by centrifugation of the extract at 5000× *g* for 10 min. The supernatant was filtered through a Whatman GF/B filter, and 3 mL of the filtrate was diluted with 12 mL of water. The Isolute columns were conditioned with 2 mL of ACN, followed by 2 mL of water. Diluted supernatant (5 mL) was loaded onto the column, which was then washed with 2 mL of water, followed by 2 mL of 10% ACN (*v*/*v*). Air was passed through the column for a few seconds. Analytes were eluted with 2 mL of ACN (0.1% formic acid, *v*/*v*) and 4 mL of MeOH. The eluate was evaporated under N_2_ at 50 °C, and the residue was reconstituted with 50% MeOH (0.1% formic acid, *v*/*v*).

#### 3.3.2. Method B

Sample preparation with a QuEChERS kit (containing 4 g of hydrous magnesium sulfate and 1 g of sodium chloride with ceramic homogenizer; Agilent Technologies, Palo Alto, CA) was conducted according to the Agilent application note and the method reported by Andrade et al., with some modifications [43]. The sample (2 g) was weighed into a 50 mL conical tube, water (10 mL) was added, the mixture was held for 5 min, and then 10 mL of ACN (2% formic acid, *v*/*v*) was added. The mixture was shaken for 20 min at 300 rpm, and then QuEChERS salt (4 g MgSO_4_, 1 g NaCl) was added. Next, ceramic homogenizer was added, and the mixture was shaken manually for 3 min. The extract was then centrifuged at 5000× *g* for 5 min. The supernatant was filtered through Whatman No. 4 filter paper. The filtrate (5 mL) was dried under a stream of N_2_ at 50 °C and then reconstituted with 50% MeOH.

#### 3.3.3. Method C

Samples were prepared using a Myco 6-in-1 IAC from Vicam (Milford, MA) according to the manufacturer’s protocol, with some modifications. The sample (10 g) was weighed into a high-form beaker, water (40 mL) was added, and the mixture was homogenized for 2 min at 5000 rpm (T25 Digital Ultra-Turrax; IKA-Werke, Staufen, Germany). Without removing the first extract, MeOH (60 mL) was added to the beaker, and the mixture was homogenized again for 2 min (5000 rpm). The extract was filtered through Whatman No. 4 filter paper. The filtrate (5 mL) was evaporated to approximately 2 mL, and then PBS (5 mL) was added. The extract/PBS mixture (7 mL) was loaded onto an IAC, which was washed with water (10 mL). The column was dried by passing air through for a few seconds. MeOH (1.5 mL) was applied to the column and eluted into a test tube, a second portion of MeOH (1.5 mL) was applied, and the bottom of the IAC was closed. After 5 min, the eluate was allowed to flow again. We passed 2 mL of water through the column into the same test tube. Subsequently, the column was dried by passing air through it for a few seconds. The eluent was evaporated under a stream of N_2_ at 40 °C and the residue was reconstituted in 50% MeOH (0.2 mL, *v*/*v*).

Finally, method C (with an IAC) was selected for further optimization because of its potential to clean up and analyze mycotoxins in a doenjang matrix simultaneously. The optimized method comprised the following steps. A total of 10 g of sample was extracted with 1 g of sodium chloride. The extraction and filtration steps were as described above. Filtered extract (4 mL) was diluted with 16 mL of PBS and then filtered, once again, through a Whatman GF/A filter. A 20 mL volume of the GF/A filtrate was loaded onto an IAC, followed by washing of the column with 4 mL of water. The mycotoxins were eluted from the IAC using 2 mL of MeOH containing 0.2% formic acid (*v*/*v*) and a further 3 mL of MeOH containing 0.2% formic acid (*v*/*v*). The bottom of the IAC was closed. The column was held closed for 5 min and the eluate was allowed to flow. The final residue was reconstituted in 0.5 mL of 50% MeOH (*v*/*v*).

### 3.4. LC-MS/MS Conditions

Detection and quantification were performed using a Thermo Accela UHPLC system (Thermo Fisher Scientific, San Jose, CA) coupled with a Velos Pro mass spectrometer (Thermo Fisher Scientific, Waltham, MA) equipped with positive and negative ESI modes. Analysis was conducted with a Waters Xbridge C_18_ column (2.1 mm × 100 mm, 3.4 μm; Waters, Milford, MA). Separation was conducted over a period of 20 min using a flow rate of 0.2 mL/min: 90% solvent A (0.1% formic acid in water), 10% solvent B (0.1% formic acid in MeOH) for 3 min to reach equilibrium. Solvent A was first applied in a linear gradient elution system from 90% to 5% within 13 min and then held to 16 min. Solvent A was then changed 90% to 16.1 min and the column was re-equilibrated to 20 min. The column was equilibrated for 3 min prior to each analysis. The column oven temperature was held at 35 °C. The collision-induced dissociation energy was 35%. Helium was used as the collision gas. Spray voltages were 5.0 and –5.0 kV for the positive and negative modes, respectively. Source heater and capillary temperatures were set at 250 °C and 275 °C, respectively. Sheath gas, auxiliary gas, and sweep gas (N_2_) were set at 35, 5, and 5 arbitrary units, respectively. The m/z range of 50–2000 was fully scanned. Data analysis was performed using Thermo Xcalibur Qual Browser 2.0 software.

### 3.5. Method Validation

Standard curves of each mycotoxin were evaluated by the coefficient of determination (*R*^2^) of six-point matrix-matched calibration curves that were constructed by plotting the peak area (signal intensity). The limit of detection (LOD) and limit of quantification (LOQ) were determined using the slope of the calibration curve (*S*) and the standard deviation (SD) of the area from the lowest concentration in the calibration curve. LOD and LOQ were calculated from the following equations: LOD = 3.3 × SD/*S* and LOQ = 10 × SD/*S*. The recovery was represented by recovery experiments according to the following equation: analyzed concentration of spiked samples calculated from matrix-matched standard/spiking concentration × 100. Precision was calculated as the relative standard deviation of the replicated recovery experiments. The intra- and inter-day recovery and precision were measured at spiking levels of 2×, 5×, 10×, and 20 × LOQ concentrations.

A blank doenjang sample free from 20 mycotoxins was spiked with each mycotoxin before sample pretreatment. To evaluate the trueness of the method, a CRM of doenjang powder for OTA (49.5 ± 1.17 μg/kg) from the KRISS was used. The trueness was calculated by dividing the concentration determined by the certified value and multiplying by 100 in consideration of recovery.

To calculate the matrix effects, six concentrations of each mycotoxin standard were added to both the blank matrix after sample preparation and the reference solvent (50% MeOH, *v*/*v*). These two sets of solutions were directly injected to the LC-MS/MS system, 10 times each. Signal suppression or enhancement (SSE) was calculated using the following equation: (slope of the matrix-matched calibration curve/slope of the solvent-based calibration curve) × 100.

The measurement uncertainty was quantified according to the EURACHEM/Cooperation on International Traceability in Analytical Chemistry (CITAC) Guide—Quantifying Uncertainty in Analytical Measurement (2012) [44]. Balances, volumetric measuring devices, reference materials, linear calibration curve interpolation, and instrumental factors were considered sources of uncertainty. Combined standard uncertainty was calculated as the positive square root of the total variance obtained by combining all the uncertainty components. The expanded measurement of uncertainty was calculated using a standard coverage factor of two for an approximate confidence level of 95%.

## Figures and Tables

**Figure 1 toxins-11-00594-f001:**
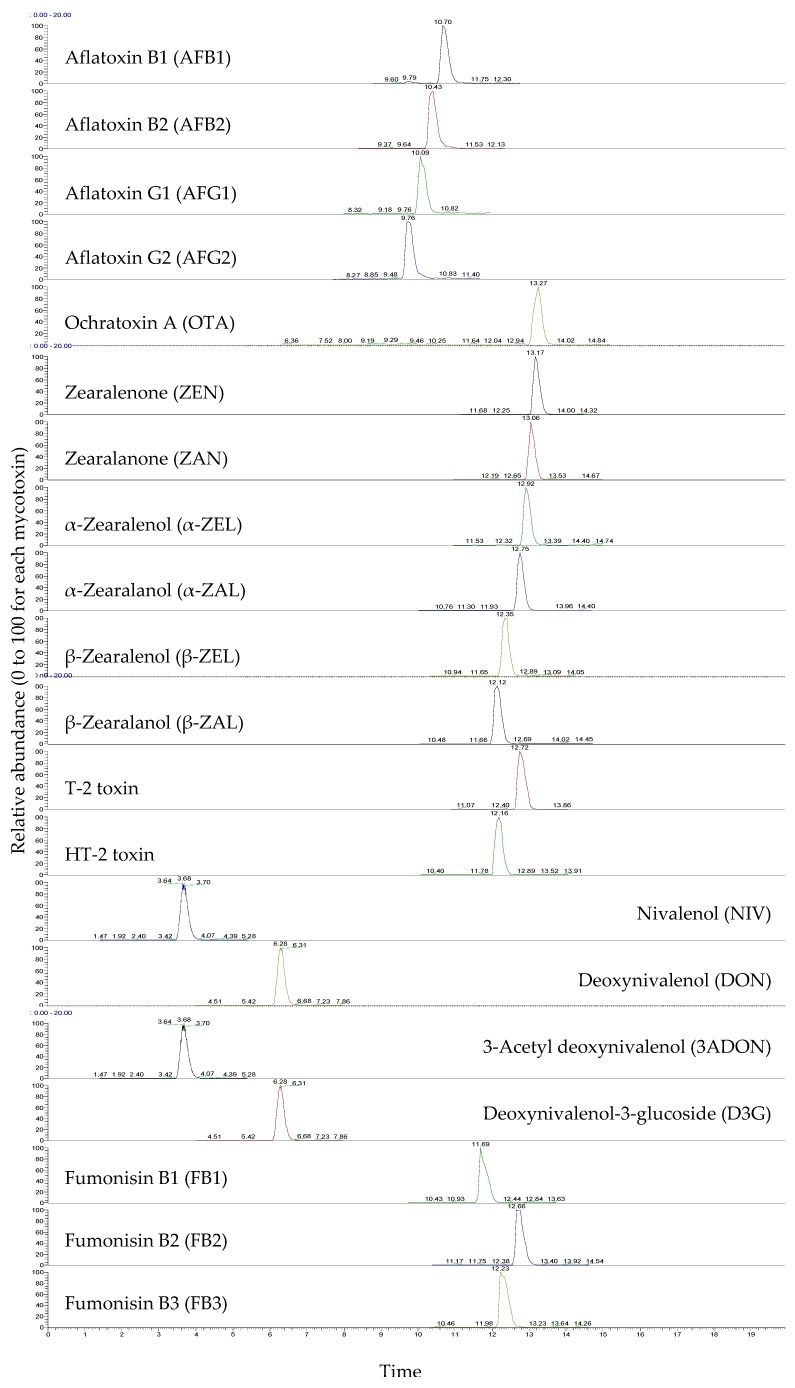
LC-MS/MS Chromatograms of the mycotoxin analytes. Concentrations were as follows: 20 μg/kg for AFB1, AFB2, AFG1, AFG2, and OTA, and 1000 μg/kg for ZEN, ZAN, α-ZEL, α-ZAL, β-ZEL, β-ZAL, T-2, HT-2, NIV, DON, 3ADON, D3G, FB1, FB2, and FB3.

**Figure 2 toxins-11-00594-f002:**
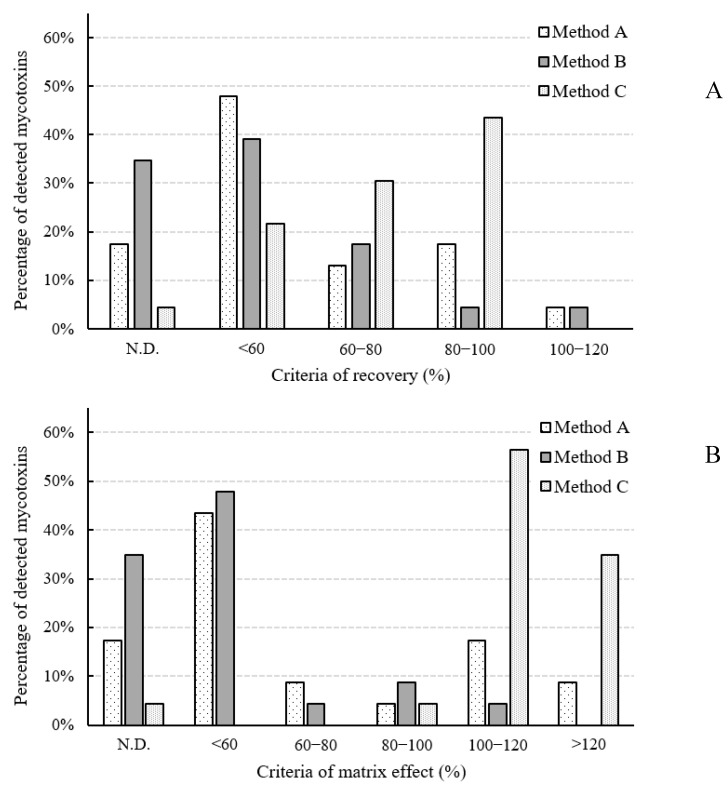
Effects of the various cleanup methods on recoveries (**A**) and matrix effects (**B**).

**Figure 3 toxins-11-00594-f003:**
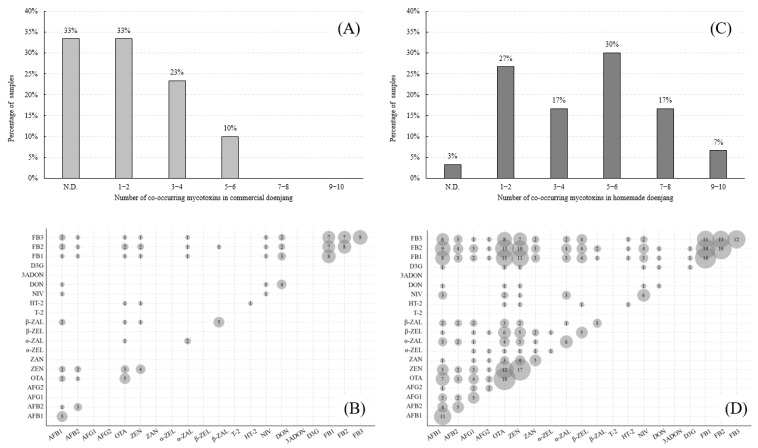
Co-occurrence and prevalence of mycotoxins in commercial (**A**,**B**) and homemade (**C**,**D**) doenjang.

**Table 1 toxins-11-00594-t001:** Optimized LC-MS/MS parameters.

Mycotoxin	Molar Mass (g/mol)	Precursor Ion	Molecular Ion	Product Ion		Retention Time (min)
				Quantifier	Qualifier	
AFB1	312.277	313.2	[M + H]^+^	285.0	284.0	10.8
AFB2	314.293	315.2	[M + H]^+^	287.0	297.0	10.4
AFG1	328.276	329.2	[M + H]^+^	311.0	301.0	10.2
AFG2	330.292	331.2	[M + H]^+^	313.0	303.0	9.8
OTA	403.815	404.8	[M + H]^+^	358.0	387.0	13.3
ZEN	318.369	319.3	[M + H]^+^	301.0	283.0	13.2
ZAN	320.385	321.0	[M + H]^+^	303.1	277.1	13.1
α-ZEL	320.385	321.0	[M + H]^+^	285.1	259.1	12.9
α-ZAL	322.401	305.0	[M + H − H_2_O]^+^	287.1	277.1	12.8
β-ZEL	320.385	303.0	[M + H − H_2_O]^+^	285.1	267.1	12.4
β-ZAL	322.401	305.0	[M + H − H_2_O]^+^	287.1	277.0	12.1
T-2	466.527	489.5	[M + Na]^+^	387.09	327.09	12.7
HT-2	424.49	447.4	[M + Na]^+^	345.0	285.0	12.2
NIV	312.318	357.2	[M + HCOO^-^]^-^	281.0	311.0	3.4
DON	296.319	341.3	[M + HCOO^-^]^-^	265.0	295.0	6.1
3ADON	338.356	339.3	[M + H]^+^	231.0	279.0	9.4
D3G	458.46	503.4	[M + HCOO^-^]^-^	457.1	427.1	6.6
FB1	721.838	722.8	[M + H]^+^	704.2	705.2	11.8
FB2	705.839	706.8	[M + H]^+^	688.3	530.3	12.8
FB3	705.839	706.8	[M + H]^+^	688.3	530.3	12.4

**Table 2 toxins-11-00594-t002:** Calibration curve, linearity, LOD, LOQ, and %SSE of the optimized method.

Mycotoxin	Calibration Curve			*R* ^2^	LOD (μg/kg)	LOQ (μg/kg)	SSE (%)
	Calibration Range (μg/kg)	Slope	Intercept				
AFB1	0.5−20	10,644.3	−142.4	0.9999	0.08	0.26	89.7
AFB2	0.5−20	8025.6	−834.4	0.9998	0.14	0.43	94.6
AFG1	0.5−20	8873.9	394.8	0.9999	0.10	0.30	100.8
AFG2	0.5−20	6878.5	−722.1	0.9999	0.15	0.45	98.7
OTA	0.5−20	20,887.2	8984.6	0.9998	0.06	0.17	87.0
ZEN	25−1000	4638.2	−5420.4	0.9999	4.68	14.18	96.7
ZAN	25−1000	11,972.7	−8964.7	1.0000	1.66	5.03	95.5
α-ZEL	25−1000	700.0	−716.6	0.9999	1.47	4.44	94.2
α-ZAL	25−1000	2098.0	2773.7	0.9998	2.29	6.94	95.9
β-ZEL	25−1000	3649.0	−12,582.2	1.0000	1.72	5.23	99.5
β-ZAL	25−1000	2042.0	8338.2	0.9997	2.13	6.45	101.9
T-2	25−1000	23,047.8	411,783.3	0.9995	2.55	7.73	103.8
HT-2	25−1000	18,293.4	257,935.3	0.9994	1.86	5.65	93.1
NIV	25−1000	1423.3	−6891.8	0.9999	3.21	9.74	98.2
DON	25−1000	941.3	−3171.0	1.0000	4.37	13.24	98.0
3ADON	25−1000	821.9	−5844.9	0.9999	0.9	2.73	100.4
D3G	25−1000	6246.2	−12,849.3	0.9999	2.72	8.24	100.3
FB1	25−1000	22,786.0	−127,037.2	0.9998	2.57	7.78	98.1
FB2	25−1000	24,992.8	−211,390.0	1.0000	2.16	6.56	87.1
FB3	25−1000	19,609.8	−21,763.0	1.0000	2.22	6.72	86.5

**Table 3 toxins-11-00594-t003:** Validation parameters: intra-/inter-day recovery and precision.

Mycotoxin	%Recovery (%RSD)															
	Intra-Day (*n* = 9)								Inter-Day (*n* = 9)							
	2 × LOQ *		5 × LOQ		10 × LOQ		20 × LOQ		2 × LOQ		5 × LOQ		10 × LOQ		20 × LOQ	
AFB1	98.4	(5.9)	97.8	(2.7)	97.5	(4.3)	101.5	(3.1)	96.5	(9.1)	93.2	(4.8)	94.7	(4.9)	97.8	(5.6)
AFB2	100.5	(4.7)	97.0	(3.9)	99.1	(2.2)	101.1	(2.3)	100.6	(3.0)	95.8	(2.8)	97.6	(3.7)	99.1	(3.8)
AFG1	100.0	(4.0)	95.6	(3.9)	95.4	(2.8)	95.7	(2.8)	94.7	(5.3)	94.7	(4.3)	93.0	(5.1)	93.0	(5.2)
AFG2	96.6	(2.2)	94.0	(2.0)	95.1	(2.2)	96.3	(4.4)	95.7	(6.4)	92.3	(5.5)	93.4	(5.3)	89.6	(9.2)
OTA	80.9	(7.2)	76.1	(5.3)	83.0	(6.5)	79.9	(6.1)	86.0	(5.0)	71.7	(2.7)	76.9	(6.4)	74.6	(3.3)
ZEN	89.4	(3.6)	85.7	(5.0)	83.1	(7.3)	84.1	(5.1)	90.8	(5.4)	83.3	(5.1)	83.8	(8.7)	85.2	(6.5)
ZAN	99.3	(2.2)	85.0	(3.1)	79.3	(6.8)	77.6	(4.8)	100.2	(2.2)	85.1	(3.7)	80.3	(6.8)	80.0	(4.8)
α-ZEL	104.5	(3.3)	93.4	(5.0)	88.6	(6.2)	85.1	(4.6)	103.6	(4.0)	92.0	(4.5)	87.9	(9.1)	86.7	(5.8)
α-ZAL	89.4	(8.2)	85.8	(11.3)	88.0	(6.1)	90.8	(4.2)	88.0	(7.5)	85.7	(8.1)	87.0	(8.7)	90.8	(6.3)
β-ZEL	105.6	(3.2)	89.4	(3.9)	83.5	(7.4)	82.0	(4.0)	109.6	(3.1)	91.0	(3.3)	85.0	(7.3)	84.1	(3.6)
β-ZAL	92.7	(5.0)	90.1	(4.8)	87.9	(6.6)	90.2	(4.5)	93.6	(5.3)	90.5	(5.8)	89.7	(6.5)	91.6	(4.4)
T-2	75.7	(16.2)	92.0	(10.4)	96.0	(12.6)	94.7	(12.6)	75.0	(4.5)	91.9	(6.0)	96.4	(8.2)	97.6	(6.6)
HT-2	79.2	(5.3)	74.3	(13.7)	76.6	(12.5)	83.0	(11.5)	86.8	(5.8)	72.6	(13.2)	78.9	(12.7)	82.1	(15.2)
NIV	95.2	(15.2)	89.5	(7.4)	89.5	(12.6)	84.3	(8.9)	112.3	(5.9)	97.1	(7.6)	86.7	(13.4)	85.5	(8.1)
DON	100.3	(6.6)	94.9	(6.3)	95.5	(8.2)	89.7	(6.6)	109.0	(6.4)	100.1	(6.5)	90.6	(9.7)	89.6	(5.1)
3ADON	103.2	(4.0)	91.5	(6.3)	90.5	(7.4)	87.0	(10.0)	107.2	(1.8)	95.1	(4.9)	96.1	(3.8)	93.6	(5.1)
D3G	95.2	(11.7)	93.2	(5.2)	93.2	(13.0)	85.8	(8.8)	106.4	(6.4)	95.8	(5.1)	88.1	(13.0)	85.3	(8.0)
FB1	94.3	(11.0)	99.5	(8.0)	95.1	(5.1)	83.8	(6.7)	93.2	(8.2)	103.5	(7.4)	92.3	(13.2)	90.3	(7.9)
FB2	100.3	(4.7)	87.6	(11.5)	89.1	(6.9)	81.1	(9.6)	103.5	(13.5)	88.9	(11.8)	86.5	(12.8)	84.8	(10.0)
FB3	96.6	(4.1)	87.0	(11.2)	91.2	(6.8)	82.2	(5.8)	97.6	(11.7)	88.6	(9.4)	88.8	(12.9)	84.2	(10.3)

* AFs and OTA: 1 μg/kg (2 × LOQ), 2 μg/kg (5 × LOQ), 3 μg/kg (10 × LOQ), 6 μg/kg (20 × LOQ); other mycotoxins: 15 μg/kg (2 × LOQ), 37 μg/kg (5 × LOQ), 74 μg/kg (10 × LOQ), 148 μg/kg (20 × LOQ).

**Table 4 toxins-11-00594-t004:** Expanded uncertainty of each mycotoxin.

Mycotoxin	Spiking Level (μg/kg)	Expanded Uncertainty	Uncertainty/Result (%)	Mycotoxin	Spiking Level (μg/kg)	Expanded Uncertainty	Uncertainty/Result (%)
AFB1	1	0.23	22.7	β-ZAL	15	1.18	15.8
	2	0.31	15.4		37	3.28	17.7
	3	0.73	14.6		74	7.53	20.3
	6	1.26	12.6		148	12.93	17.5
AFB2	1	0.20	19.8	T-2	15	2.38	31.7
	2	0.31	15.3		37	2.40	13.0
	3	0.69	13.8		74	2.92	7.9
	6	1.32	13.2		148	6.02	8.1
AFG1	1	0.20	20.3	HT-2	15	2.86	19.1
	2	0.31	15.5		37	7.23	39.1
	3	0.72	14.4		74	9.85	26.6
	6	1.39	13.9		148	18.92	25.6
AFG2	1	0.20	19.6	NIV	15	1.10	14.7
	2	0.32	15.8		37	1.76	9.5
	3	0.72	14.3		74	4.33	11.7
	6	1.35	13.5		148	11.86	16.0
OTA	1	0.41	41.3	DON	15	0.69	9.2
	2	0.83	41.6		37	1.89	10.2
	3	1.42	28.4		74	3.88	10.5
	6	3.11	31.1		148	10.49	14.2
ZEN	15	2.02	13.4	3ADON	15	0.85	11.4
	37	7.21	19.5		37	2.18	11.8
	74	16.66	22.5		74	4.30	11.6
	148	29.22	19.7		148	10.16	13.7
ZAN	15	1.95	13.0	D3G	15	0.78	10.3
	37	8.56	23.1		37	1.50	8.1
	74	22.02	29.8		74	3.09	8.3
	148	47.40	32.0		148	11.45	15.5
α-ZEL	15	2.01	13.4	FB1	15	1.00	6.7
	37	5.84	15.8		37	1.22	6.6
	74	14.41	19.5		74	2.40	6.5
	148	33.21	22.4		148	10.74	14.5
α-ZAL	15	2.66	17.8	FB2	15	0.50	6.7
	37	7.80	21.1		37	2.86	15.4
	74	14.86	20.1		74	4.80	13.0
	148	25.70	17.4		148	17.58	23.8
β-ZEL	15	2.25	15.0	FB3	15	0.53	7.0
	37	6.57	17.7		37	2.77	15.0
	74	17.36	23.5		74	3.28	8.9
	148	38.32	25.9		148	16.22	21.9

**Table 5 toxins-11-00594-t005:** Application to commercial and homemade doenjang samples.

Mycotoxin	Commercial *doenjang* (μg/kg), *n* = 30				Homemade *doenjang* (μg/kg), *n* = 30			
	Incidence ^a^	Mean (Positive Mean)	Median	Range	Incidence ^a^	Mean (Positive Mean)	Median	Range
AFB1 *	5/30	0.07 (0.40)	0.33	0.11−0.96	11/30	0.54 (1.48)	0.65	0.19−4.45
AFB2	3/30	0.08 (0.77)	0.77	0.70−0.84	5/30	0.14 (0.82)	0.87	0.62−0.98
AFG1 *	0/30	- ^b^	-	-	5/30	0.10 (0.60)	0.57	0.49−0.77
AFG2	0/30	-	-	-	2/30	0.03 (0.47)	0.47	0.21−0.73
OTA	5/30	0.80 (4.80)	0.19	0.16−23.27	18/30	3.05 (5.08)	3.78	0.20−19.68
ZEN **	4/30	2.79 (20.90)	17.47	5.16−43.48	17/30	17.82 (31.45)	19.40	5.86−85.50
ZAN *	0/30	-	-	-	5/30	0.94 (5.67)	6.18	3.28−7.11
α-ZEL	0/30	-	-	-	1/30	0.28 (8.29)	8.29	8.29
α-ZAL	2/30	0.34 (5.17)	5.17	5.13−5.21	6/30	1.04 (5.21)	5.01	4.05−7.18
β-ZEL *	0/30	-	-	-	5/30	1.56 (9.39)	9.91	7.38−11.56
β-ZAL	5/30	0.93 (5.56)	5.64	4.67−6.19	3/30	1.66 (16.56)	15.59	8.40−25.70
T-2	0/30	-	-	-	0/30	-	-	-
HT-2	1/30	0.88 (26.52)	26.52	26.52	1/30	0.06 (1.69)	1.69	1.69
NIV	1/30	0.26 (7.67)	7.67	7.67	6/30	0.89 (4.43)	4.31	3.27−6.03
DON	4/30	1.72 (12.88)	14.25	4.78−18.25	1/30	0.51 (15.35)	15.35	15.35
3ADON	0/30	-	-	-	0/30	-	-	-
D3G	0/30	-	-	-	1/30	0.09 (2.72)	2.72	2.72
FB1 **	8/30	1.50 (5.61)	6.64	2.21−8.13	16/30	5.04 (9.44)	9.39	5.52−17.26
FB2 *	8/30	1.57 (5.90)	6.20	1.85−9.11	16/30	7.00 (13.33)	6.03	3.86−56.19
FB3	9/30	1.68 (5.60)	6.10	2.48−9.22	12/30	2.00 (5.00)	4.96	2.37−9.36

^a^: number of positive samples/total number of samples; ^b^: not detected (<LOD); difference between commercial and homemade products (* *p* < 0.05); difference between commercial and homemade products (** *p* < 0.01).

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
