# Peer review of "Simultaneous Determination of Twenty Mycotoxins in the Korean Soybean Paste Doenjang by LC-MS/MS with Immunoaffinity Cleanup"

_toxins, 2019, doi:10.3390/toxins11100594_

Round 1
Reviewer 1 Report
See attached document

Author Response
Responses to the reviewer 1 comments
Point 1: The authors are using incorrect terminology. What they describe as accuracy throughout the manuscript should be named trueness. Accuracy is a combination of trueness and precision (as stated in [36] and more recently the "International Vocabulary of Metrology"). Reproducibility (line 175) can only be determined through a interlaboratory comparison. What the authors describe is either repeatability or intermediate precision (ISO 5725). Revise.
Response 1: Thank you for your suggestion. As you suggested, we modified all the words “accuracy” to “recovery” in the revised manuscript. In our original manuscript, we only used the term ‘reproducibility’ to describe general features of the matrix effect. Throughout our manuscript, we used a term ‘repeatability” to describe the %RSD of the recovery tests.
Point 2: (L66) ESI suffers from matrix effect it does not cause them. And matrix-matched calibration does not change anything about the matrix effects themselves but corrects for them. IDMS would need to be mentioned here, too. Revise.
Response 2: We modified expression “ESI…can cause…” to “ESI…can be affected by…” (Line 69-72).
Point 3: (L69) LC-MS/MS certainly does not need improvement of selectivity through a clean-up which is why many "dilute & shoot" methods work well with LC-MS/MS. Revise.
Response 3: This sentence was originally intended to explain the general characteristics of clean-up process. We modified the expression “is required” to “is important” because it can cause confusion (Line 73).
Point 4: (L71) IAC is certainly not "...quick, easy, cheap...". Revise.
Response 4: This sentence did not mean IAC method is quick, easy, cheap (Line 74). This caused because our English was not very functional. In the revised manuscript, we modified this sentence. The modified sentence reads “In this regard, the application of IAC in mycotoxin analysis is a widely used method in recent years, and a fast, easy, inexpensive, effective, robust and safe (QuEChERS) strategy is also used because of its high selectivity and ability to reduce matrix effects” (Line 74-77).
Point 5: (L89ff) The descriptions of the measured precursor ions are partly wrong. "[M-OH]+” does not exist. Correctly it should read [M+H-H2O]+ because water fragments off the protonated ZEN metabolite molecule. Also "[M+H2O]+" is wrong for the T-2 toxin ion. Water is a neutral which combined with the neutral T-2 cannot result in a positively charged ion. What the authors most likely have observed is [M+NH4]+, the ammonium adduct of T-2 toxin. Revise.
Response 5: Thank you for your careful review, and sorry for our carelessness. In response to reviewer’s indication, we changed descriptions of the precursor ions of ZEN metabolite molecule to [M+H-H2O] + (Line 93-94). In addition, we found that the precursor ions of T-2 used in this study was [M + Na]+, same as the molecular form of precursor ion of HT-2 toxin (Line 94). Accordingly, we modified the molecular ion of T-2 in the Table 1.
Point 6: (L99) What is the shading in Fig 1 supposed to depict? Remove.
Response 6: In response to reviewer’s indication, we removed the shading in Fig 1 in the revised manuscript (Line 103). In addition, the first letter of some mycotoxins, which were shown in lowercase in Figure 1, was capitalized.
Point 7: (L153) What the authors describe here would point to unspecific binding of FBs to the IAC column. That defeats the purpose of IAC. Needs to be clarified.
Response 7: What was described in this manuscript (Line 155−158) is not that FBs were unspecifically bound to antibodies in IAC. As you know, multifunctional IAC are packed with activated solid phase bound to specific antibodies for given mycotoxins. When the extract passes through the column, multi-mycotoxins bind selectively to its specific antibodies, while other matrix components should be removed by a washing step. We found that unlike other mycotoxins, FBs were eluted in the washing water as they are soluble in water. So we focused to minimize the loss of FBs in the washing step. To clarify this, we added “to minimize loss of FBs in the washing step” in the revised manuscript (Line 158-159).
Point 8: (L160ff) The wording of this paragraph is more or less identical to L104-111. Remove.
Response 8: Sorry to our mistake. We removed duplicate paragraph.
Point 9: (L168) The coefficient of determination (R squared) is not an indicator of linearity but of goodness-of-fit. To prove linearity a Mandel linearity test or better yet a residual plot is to be used. Revise.
Response 9: Thank you for your suggestion. As you mentioned, the coefficient of determination (R squared) is not an indicator of linearity but of goodness-of-fit. However, R squared close to 1 is still considered by some authors’ sufficient evidence to conclude that the calibration curve is linear. Although we used R squared as an indicative of linearity, we added an expression “although R2 is an indicator of goodness-of-fit rather than linearity” to convey more accurate meanings in the revised manuscript (Line 166).
Point 10: (L200) Reference [39] is lacking document source details.
Response 10: In response to reviewer’s indication, we added the page number on reference [40].
Point 11: (L277) "...standard solutions..." would not need to be "...dissolved in..." ACN. Revise.
Response 11: Thank you for your comments. It was modified “…standard solutions… dissolved in…” to “…standards… dissolved in…” (Line 279).
Point 12: (L348ff) The gradient described here has a duration of 38 min with three equilibrations of 3 min each. Which 20 min were used for the separation?
Response 12: In response to reviewer’s indication, we changed the explanation about the gradient condition (Line 353−356).
Point 13: (L353) The normalised collision energy of a Velos Pro MS is not expressed in eV but in %. Revise.
Response 13: Sorry to our mistake. We modified eV to % (Line 357).
Point 14: (L369) Since mycotoxins contamination seems to be so prevalent where was the analyte-free doenjang material obtained. How were the matrix-matched calibration solutions prepared? LOD/LOQ as described here only state the instrument performance since they apparently did not include extraction effects. That needs to be clarified.
Response 14: Thank you for your comments. First of all, as reviewer said, it was hard to obtain the doenjang sample free from twenty mycotoxins. Even if it is not mentioned in the manuscript, we analyzed dozens of doenjang samples and obtained blank doenjang samples free from multi-mycotoxins. In the revised manuscript, we added “free from 20 mycotoxins” (Line 373).
The LOD and LOQ that calculated in this study state the instrument performance. However, we often seen things expressed in this way in published reports. To compare our results with previously reported results, we calculated LOD and LOQ by using calibration curve.
Point 15: (L373) The description of how matrix effects were quantified is confusing. It seems to
describe three sets of solutions of which only two were used for calculations.
Response 15: The matrix effect was calculated as the ratio of the slope of the matrix-matched calibration curve and the solid-based calibration curve. In order to clearly describe the matrix effect, we changed from "final reconstitution solvent” to “the reference solvent”, from “Six concentration standards” to “These two sets of solutions” and from “the liquid standard” to “solvent-based” in the revised manuscript (Line 380-383).
Point 16: (L386) I was not able to find Figure S1
Response 16: Sorry to our mistake. We uploaded the supplementary materials together.
Reviewer 2 Report
The manuscript describe the optimization and validation of LC-MS/MS method of analysis of several mycotoxins in fermented soybean product.
The paper is clearly presented, the aim is clear and the data are correctly reported.
Some weakness are related to the introduction part that need to be implemented comparing the actual procedures for similar determination and also in the discussion of the results authors should more in detail express to the readers the innovation and the novelty of their findings.
Introduction and references to previous work need to be implemented.
Comment related to the use of the proposed methods as alternative of other methods need also to be considered and discussed.
Overall the manuscript can be of interest and as presented need revision but mandatory is to improve introduction and discussion.
Author Response
Responses to the reviewer 2 comments
Point 1: Overall the manuscript can be of interest and as presented need revision but mandatory is to improve introduction and discussion.
Response 1: Thank you for your comments. We have carefully reviewed our paper again. In response to reviewer’s suggestion, some additional sentences have been added to the introduction and discussion.
In introduction part, we added a sentence to describe the difficulty of applying SPE or ‘dilute & shoot’ to the doenjang matrix (Line 44−47). The added sentence reads “For quantification of trace levels of mycotoxin in soybean paste matrix, analysis without the clean-up process or using solid phase extraction (SPE) that packed with general C18 or ion pair materials resulted in insufficient removal of salts and other polar components in matrix.[4]”. Accordingly, a reference was added in the revised manuscript. In addition, we mentioned “isotope dilution mass spectrometry technique” (Line 71-72).
In discussion part, we added a sentence to describe the novelty of our study (Line 265−268). The added sentence reads “Furthermore, to the best of our knowledge, this study is the first attempt to simultaneously determine up to 20 mycotoxins after sample cleanup using multifunctional IACs in soybean fermented foods”.
In addition to the responses shown above, we corrected from “A. oryza or A. sojae” to “A. oryza or A. sojae” (L262), “R2 “ to “R2” (Line 11) and from “fumonisins” to “FBs” (Line 162) in the revised manuscript. In addition, the first letter of some mycotoxins, which were shown in lowercase in Figure 1, was capitalized.
I would like to thank for reviewers’ comments and suggestions again. We think that the quality of our manuscript has been highly improved by the process of review and revision. I hope we have addressed all points to your and reviewer’s satisfaction.